# Derivation and Validation of Clinical Prediction Models for Rapid Risk Stratification for Time-Sensitive Management for Acute Heart Failure

**DOI:** 10.3390/jcm9113394

**Published:** 2020-10-23

**Authors:** Yasuyuki Shiraishi, Shun Kohsaka, Takayuki Abe, Toshiyuki Nagai, Ayumi Goda, Yosuke Nishihata, Yuji Nagatomo, Mike Saji, Yuichi Toyosaki, Makoto Takei, Takeshi Kitai, Takashi Kohno, Keiichi Fukuda, Yuya Matsue, Toshihisa Anzai, Tsutomu Yoshikawa

**Affiliations:** 1Department of Cardiology, Keio University School of Medicine, Tokyo 160-8582, Japan; yasshiraishi@keio.jp (Y.S.); kfukuda@a2.keio.jp (K.F.); 2Biostatistics, Clinical and Translational Research Center, Keio University School of Medicine, Tokyo 160-8582, Japan; abe.t@keio.jp; 3School of Data Science, Yokohama City University, Yokohama 236-0027, Japan; 4Department of Cardiovascular Medicine, National Cerebral and Cardiovascular Center, Osaka 565-8565, Japan; tnagai@huhp.hokudai.ac.jp (T.N.); toshianzai@gmail.com (T.A.); 5Department of Cardiovascular Medicine, Faculty of Medicine and Graduate School of Medicine Hokkaido University, Sapporo 060-8638, Japan; 6Department of Cardiovascular Medicine, Kyorin University School of Medicine, Tokyo 181-8611, Japan; ayumix34@yahoo.co.jp (A.G.); kohno.a2@keio.jp (T.K.); 7Department of Cardiology, St. Luke’s International Hospital, Tokyo 104-8560, Japan; hatasuke@luke.ac.jp; 8Department of Cardiology, National Defense Medical College Hospital, Tokorozawa 359-8513, Japan; y.nagatomo1111@gmail.com; 9Department of Cardiology, Sakakibara Heart Institute, Tokyo 183-0003, Japan; mikesaji8@gmail.com (M.S.); tyoshi@shi.heart.or.jp (T.Y.); 10Department of Cardiology, Saitama Medical University International Medical Center, Saitama 350-1298, Japan; toyoyuki6465@yahoo.co.jp; 11Department of Cardiology, Tokyo Saiseikai Central Hospital, Tokyo 108-0073, Japan; makoto_tk@hotmail.com; 12Department of Cardiovascular Medicine, Kobe City Medical Center General Hospital, Kobe 650-0047, Japan; t-kitai@kcho.jp; 13Department of Cardiovascular Biology and Medicine, Juntendo University Graduate School of Medicine, Tokyo 113-8421, Japan; yuya8950@gmail.com

**Keywords:** acute heart failure, prediction model, in-hospital mortality, discrimination, calibration, validation

## Abstract

Early and rapid risk stratification of patients with acute heart failure (AHF) is crucial for appropriate patient triage and outcome improvements. We aimed to develop an easy-to-use, in-hospital mortality risk prediction tool based on data collected from AHF patients at their initial presentation. Consecutive patients’ data pertaining to 2006–2017 were extracted from the West Tokyo Heart Failure (WET-HF) and National Cerebral and Cardiovascular Center Acute Decompensated Heart Failure (NaDEF) registries (*n* = 4351). Risk model development involved stepwise logistic regression analysis and prospective validation using data pertaining to 2014–2015 in the Registry Focused on Very Early Presentation and Treatment in Emergency Department of Acute Heart Failure Syndrome (REALITY-AHF) (*n* = 1682). The final model included data describing six in-hospital mortality risk predictors, namely, age, systolic blood pressure, blood urea nitrogen, serum sodium, albumin, and natriuretic peptide (SOB-ASAP score), available at the time of initial triage. The model showed excellent discrimination (c-statistic = 0.82) and good agreement between predicted and observed mortality rates. The model enabled the stratification of the mortality rates across sixths (from 14.5% to <1%). When assigned a point for each associated factor, the integer score’s discrimination was similar (c-statistic = 0.82) with good calibration across the patients with various risk profiles. The models’ performance was retained in the independent validation dataset. Promptly determining in-hospital mortality risks is achievable in the first few hours of presentation; they correlate strongly with mortality among AHF patients, potentially facilitating clinical decision-making.

## 1. Introduction

Prevalence of heart failure (HF) is increasing throughout the world, and HF admissions are associated with high morbidity and mortality rates, and high costs [1,2,3]. It is widely recognized that acute heart failure (AHF) hospitalization incurs most of the health care expenditure for HF [4], yet novel effective therapeutic strategies that could improve patients’ long-term prognoses have not been identified [3,5,6]. Hence, risk stratification is a key element that facilitates discrimination between low- and high-risk patients and tailors HF management. While low-risk patients may not benefit from advanced monitoring and mechanical circulatory support, these management approaches may be futile for extremely high-risk patients. Another potential advantage of risk scoring systems is their applicability to a guide to discharge low-risk patients from hospitals without increasing mortality.

Previously validated risk scores are frequently based on data describing different parameters, some of which are readily available when patients present during routine clinical practice [7,8,9,10], though calculated risks do not always correspond closely. Additionally, some of risk scores tend to be derived from data describing patient cohorts that were obtained many years before the models became available, and they may need to be refined as standard of clinical practice evolve.

Given these perspectives, treatment strategies based on patients’ absolute risks may be reasonable and improve AHF care [11], particularly in a climate of increasingly expensive therapies and cost-containment strategies. This study aimed to develop an easy-to-use, in-hospital mortality risk prediction tool based on readily available data from patients with AHF at their initial presentation, derived from and validated using contemporary, independent nationwide registry datasets.

## 2. Materials and Methods

### 2.1. Study Cohort and Sample

The present study evaluated data from three representative Japanese AHF registries: the West Tokyo Heart Failure (WET-HF) registry, National Cerebral and Cardiovascular Center Acute Decompensated Heart Failure (NaDEF) registry, and Registry Focused on Very Early Presentation and Treatment in Emergency Department of Acute Heart Failure Syndrome (REALITY-AHF). The design of these registries has been described previously [12,13,14]. Briefly, the WET-HF registry is a prospective multicenter cohort registry that contains data describing the clinical backgrounds and outcomes, including patient-reported outcomes, of patients hospitalized with AHF [12]. From 2006 to 2017, patients with AHF from six tertiary care hospitals were consecutively registered in the WET-HF registry. The NaDEF registry is a single-center, prospective cohort registry that includes consecutive patients who required hospitalization for AHF from 2013 to 2015 [13]. The REALITY-AHF is a prospective multicenter registry designed to facilitate evaluations of associations between the time to treatment and outcomes in patients with AHF presenting at emergency departments [14]. From 2014 to 2015, patients with AHF from 20 hospitals in all regions of Japan were consecutively registered in the REALITY-AHF.

In all of the registries, AHF is defined according to the Framingham criteria as rapid-onset HF or a change in the signs and symptoms of HF requiring urgent therapy and hospitalization [15]. Patients with B-type natriuretic peptide (BNP) levels < 100 pg/mL or N-terminal proBNP (NT-proBNP) levels < 300 pg/mL at baseline were excluded from the REALITY-AHF. Experienced cardiologists at each participating hospital diagnosed AHF clinically according to chart reviews. The WET-HF and NaDEF registries include patients with AHF who had scheduled and unscheduled hospital visits and/or were admitted, irrespective of the route to hospital admission (i.e., via emergency departments or routine clinics), whereas the REALITY-AHF only registered patients hospitalized through emergency departments. The three registries’ databases contain similar data elements that have standardized definitions. Patients presenting with acute coronary syndrome were excluded from the present study. A total of 6033 patients were registered in these datasets, and to develop the risk prediction models, patients in the WET-HF and NaDEF registries comprised the derivation cohort, and those in the REALITY-AHF registry comprised the validation cohort.

Each site’s institutional review board approved the study protocol, and the research was conducted in accordance with the principles of the Declaration of Helsinki. Written or verbal informed consent was obtained from each subject before the study began. The study was planned and performed in accordance with the Strengthening the Reporting of Observational Studies in Epidemiology guidelines.

### 2.2. Outcome and Variable Definitions

The study’s primary endpoint was in-hospital mortality. Data describing clinical variables, including the vital signs and laboratory findings, including the natriuretic peptide levels, were obtained at presentation and evaluated.

### 2.3. Statistical Analysis

Data describing the continuous variables are expressed as the means and the standard deviations (SDs) or as medians with the interquartile ranges (IQRs), and data describing the categorical variables are expressed as numbers and percentages. Statistical comparisons between the derivation and validation cohorts were performed using the unpaired t test or Mann–Whitney U test for continuous variables and the Pearson’s chi-squared test for categorical variables. Associations between the covariates and in-hospital mortality were evaluated using univariable and multivariable logistic regression models, and unadjusted and adjusted odds ratios and their 95% confidence intervals were estimated.

All patients were included in the multivariable regression analysis by means of multiple imputation, which is based on Bayesian theory and is a principled missing data method that provides valid statistical inferences under missing at random (MAR) conditions. MAR assumes that missing values can be explained using all of the observed data. The multiple imputation by chained equations (MICE) algorithm was applied, and linear regression, logistic, and discriminant imputation methods were used for the continuous, binary, and multinominal variables, respectively. All the covariates and the response variable (e.g., mortality) were included as covariates in each imputation model. There were 10 burn-in iterations between imputations, and 10 imputations were included in the MICE algorithm. Rubin’s rules were used to combine the estimates and determine their precision from analyses of multiple imputed data.

The model that was created using data from the WET-HF and NaDEF registries incorporated data describing clinically relevant variables that were available in the emergency department within 1 h and had values of *p* < 0.01 in the multivariable analysis with backward elimination, namely, age; systolic blood pressure (SBP); and blood urea nitrogen, serum sodium, albumin, and natriuretic peptide levels (i.e., BNP or NT-proBNP) (*SOB-ASAP score*: SO, serum SOdium; B, Blood urea nitrogen; A, Age and Albumin; S, Systolic blood pressure; and P, natriuretic Peptide). The discriminative ability of the model was assessed using the concordance (c)-statistic. The model’s calibration performance was assessed by comparing predicted and observed probability for 6 groups. The model was then validated externally using the validation cohort from the REALITY-AHF by assessing its performance, discrimination, and calibration. Furthermore, the discriminative ability of the model was compared to the Get With The Guideline-Heart Failure (GWTG-HF) risk score [12], a validated risk model for predicting in-hospital mortality, by Delong’s test.

The model was remade using the derivation cohort to develop a risk score that predicted in-hospital mortality. A weighted integer was assigned to each independent predictor, based on the predictor’s coefficient in the regression model. For each patient, the weighted integers were summed to obtain a total risk score that ranged from 0 to 14 points. The discriminative ability of the risk score was assessed using the c-statistic, and its calibration was evaluated by comparing predicted and observed mortality rates. The final analyses were performed in February 2020, all of the tests were 2-sided, and values of *p* < 0.05 were considered statistically significant. All of the statistical analyses were performed using SAS software, version 9.4 (SAS Institute, Inc., Cary, NC, USA).

## 3. Results

### 3.1. Patients’ Characteristics

Table 1 summarizes the study population’s baseline characteristics. The derivation cohort comprised 4351 patients who were predominantly male (59.9%), their mean (SD) age, SBP, and left ventricular ejection fraction (LVEF) were 74.6 (13.1) years, 140.1 (33.6) mm Hg, and 43.5% (15.9%), respectively, and their median (IQR) BNP and NT-proBNP levels were 658 (335–1209) pg/mL and 3867 (1917–8741) pg/mL, respectively. Compared to the patients in the derivation cohort, the patients in the validation cohort (*n* = 1682) were older and had higher SBPs, heart rates, LVEFs, and BNP or NT-proBNP levels (all *p* < 0.001).

### 3.2. Model Development and Validation

Overall, 171 (3.9%) patients in the derivation cohort and 85 (5.1%) in the validation cohort died during their index hospitalizations. Table 2 presents the results from the univariable logistic regression analyses of associations between each variable and in-hospital mortality. After backward elimination involving the clinically relevant variables for which variables could be obtained within 1 h in the emergency department, the final model comprised six variables (Table 3). Regarding in-hospital mortality, the model showed excellent discrimination (c-statistic = 0.82) and calibration (Figure 1A). The model enabled the stratification of the mortality rates across sixths, from 14.5% in the highest sixth to <1% in the two lowest sixth. In the complete-case analysis without multiple imputation (*n* = 2524), the model performance was the same as the model derived from the imputation dataset. When validated externally, the model showed good discrimination for in-hospital mortality (c-statistic = 0.71), and its calibration is demonstrated in Figure 1B. Compared to the GWTG-HF risk score, the model showed a good discriminative ability, with an absolute difference in c-statistics of 0.076 (*p* < 0.001) in the derivation cohort. In the validation cohort, there was no significant difference in the model performances (−0.027, *p* = 0.361).

### 3.3. Integer Score Calculation and Validation

Table 4 shows the chart used to calculate the integer scores that were based on the same variables as those used in the model to predict in-hospital mortality. The in-hospital mortality probabilities were estimated for the patients by summing the scores assigned to each predictor’s value; the total scores ranged from 0 to 14. The integer score demonstrated excellent discrimination (c-statistic = 0.82) and calibration in the derivation cohort (Figure 2A). In the validation cohort, the integer score showed good discrimination (c-statistic = 0.70) and its calibration was demonstrated in Figure 2B.

In addition, we divided the patients in the derivation cohort into three groups based on the tertile of the integer score (1st tertile, 1 and 2nd tertile, 4). Regarding in-hospital management during the clinical course of hospitalization, inotropes and other invasive treatments were frequently used in the high-score group (Table 5). The intravenous administration of diuretics occurred more frequently in the high-score group, while no apparent trend in the use of intravenous vasodilators was observed across the groups. The length of hospital stay was longer with increasing estimated risks (median, 14 days in the low-risk score group, vs. 16 days in the intermediate-risk score group vs. 20 days in the high-risk score group; *p* < 0.001).

## 4. Discussion

Using three prospective Japanese registries, we found significant predictors of in-hospital mortality and developed risk models that could estimate in-hospital mortality among patients with AHF. The data for all the variables used in the risk models are available during the acute phase of AHF management and within the first few hours of patient presentation. The models created based on the derivation cohort showed excellent discrimination and calibration, and the application of the models to the validation cohort demonstrated good levels of discrimination and calibration for most of the patients.

Accurately discriminating between high- and extremely high-risk patients (mortality rates exceeding 10–30%) is important to determine whether further treatment is futile. Moreover, given that the current management of patients with HF involves frequent hospitalizations and incurs high costs, separating patients with AHF into low- (mortality rate less than 1%) and intermediate to high-risk groups is crucial to avoid an unnecessary hospitalization without increasing mortality. The results of our study demonstrated that approximately one-third of the patients hospitalized for AHF had a low in-hospital mortality rate of less than 1% (the patients in the 2 lowest). In the USA, the Hospital Readmissions Reduction Program (HRRP) was implemented to prevent rehospitalizations by improving care transition processes for early follow-up of hospitalized patients with HF, and analyses of the data from this program revealed noteworthy results. After implementing the HRRP, 30-day and 1-year rehospitalization rates declined, but 30-day and 1-year mortality rates increased, even after risk adjustments [16]. In addition, some studies’ findings suggested that after HRRP implementation, hospitals increased the use of observational units compared to inpatient care within 30 days of discharge [17,18,19,20]. To maintain patient safety, health insurance schemes embrace more sophisticated programs that are based on individuals’ risks and reliable systems to manage early outpatient follow-up and prevent arbitrary decision-making by physicians.

In the acute setting of AHF management, natriuretic peptides, including BNP and NT-proBNP, are ideal biomarkers that can be used to identify high-risk populations easily, because they can play key roles in risk stratification, AHF diagnoses, and evaluating treatment responses [3,5,6]. Comprehensive risk scoring tools that can be populated by data obtained within the first 1–2 h of hospital admission are needed to evaluate patients in emergency departments; these data should be described from objective indicators that are not subject to interobserver errors. In the Rapid Emergency Department Heart failure Outpatient Trial (REDHOT) study that included patients presenting to the emergency department with dyspnea, there was a large disconnect between the perceived severity of HF status (i.e., the New York Heart Association functional class) by emergency physicians and BNP levels, and a lower BNP level portend a favorable prognosis [21]. Moreover, in an emergency department setting that involved patients with AHF, risk assessments using a risk prediction model were superior to physician-estimated risk assessments, and the latter overestimated the mortality risk, thereby highlighting the need for an objective score [22]. We have previously reported that risk prediction models derived from large-scale registries in Western countries, namely, the GWTG-HF and the Meta-analysis Global Group in Chronic Heart Failure risk scores, performed modestly when they were applied to Japanese patients with AHF, and that incorporating BNP levels into the models potentially improved their prognostic abilities [12,23], which concurred with the findings from several studies from different countries [8,9]. In this study, we also compared our risk model to the GWTG-HF risk score, which indicated the superiority of our model in the derivation cohort but not the validation cohort. The risk models developed in this study included some of variables used in the GWTG-HF risk score, namely, age, SBP, BUN, and serum sodium, and new parameters, specifically, BNP or NT-proBNP and serum albumin, which are significantly associated with clinical outcomes in patients with AHF. In general, natriuretic peptides and the nutritional status, which is represented by the serum albumin level, are considered universal prognostic indicators for patients with HF; therefore, our risk models may be easy to be accepted during validation studies in other countries and regions.

The results from randomized controlled trials and observational studies have shown that AHF treatment is time-sensitive, and several studies’ findings have demonstrated ways to use existing therapies more effectively. By identifying unique subpopulations within these heterogeneous patient populations, we will be able to choose time-sensitive treatments based on patients’ individual etiologies, pathophysiologies, and risk factor profile, which will help guide the development of individual therapeutic approaches that can start promptly. The inclusion criteria used in previous trials that focused on the SBP and BNP/NT-proBNP levels were not sufficiently robust to identify patients who could benefit from early intervention [24,25,26]. More comprehensive risk prediction tools that incorporate clinical parameters and biomarkers may provide the foundation for improving the outcomes of patients with AHF, because they could help physicians to determine when medical, invasive, or palliative therapies may be most appropriate or futile. In the present study, although the relationship between absolute risk and pathophysiology may not be parallel, diuretics were administered less frequently in low-risk patients, while vasodilators were routinely used irrespective of individuals’ absolute risks. Several studies have described the value of a risk-based management approach for patients with AHF [14,27,28,29]; however, to our knowledge, a risk-based approach involving early intervention has not been assessed prospectively or in a randomized manner in the context of designing therapies to alter the natural history of AHF. Further investigations are needed in the near future that prospectively confirm the improvement of outcomes for patients with AHF using a risk-based approach.

## 5. Limitations

The databases have several inherent limitations. First, although the databases included a relatively large number of patients, the in-hospital mortality rate was low, at around 4%. The external validation of the models and comparisons between models indicated that their discrimination and calibration abilities were limited. However, our models led to the successful separation of the patients into sixths that represented different levels of risk. Second, all the patients registered in the three databases were hospitalized, and patients who were discharged from emergency departments were not included. From the perspective of Japan’s health insurance system, it is very rare for patients with AHF who present with dyspnea or other symptoms to be treated transiently in an outpatient-only setting and to be discharged directly from emergency departments. Third, the route to hospital admission may have influenced model validation. The findings from studies conducted in the USA and United Kingdom showed that about two-thirds of hospitalized patients with HF transit emergency departments, and that the characteristics of patients hospitalized via emergency departments and via other routes differed substantially [30,31]. Our models could be applied to emergency departments and, for example, to routine care clinics. Forth, the definition of AHF diagnosis was not the same criteria across the registries. Fifth, data of albumin, which were included in our models, were missing in approximately one-third of cases. However, multiple imputation and sensitivity analyses using a complete-case dataset yielded the same results. Finally, almost all of the patients who were registered in our databases were Japanese people, where several important differences in patient characteristics and treatment patterns existed between Japan and other countries. Thus, the performance of our risk models must be evaluated following their application to people of other ethnicities and in other countries.

## 6. Conclusions

Using data from large-scale registries, we found major clinical contributors to in-hospital mortality among patients hospitalized for AHF. The data describing these predictors of mortality are readily available within the first few hours in clinical practice, and the risk model that incorporates these six clinical indicators easily estimates mortality rates. Although our models should not be used in isolation for decisions about patient admission or discharge, they can aid the risk stratification of patients with AHF at the time of their initial presentation. In addition, estimating in-hospital mortality may be helpful for calibrating AHF management after urgent therapy in emergency departments. Among high-risk patients, tailored treatment that incorporates a time-sensitive approach and is based on patients’ pathophysiologies and individual risks may alter their prognoses. Further investigations are warranted to generate definitive evidence that supports this risk-based approach to the management of AHF.

## Figures and Tables

**Figure 1 jcm-09-03394-f001:**
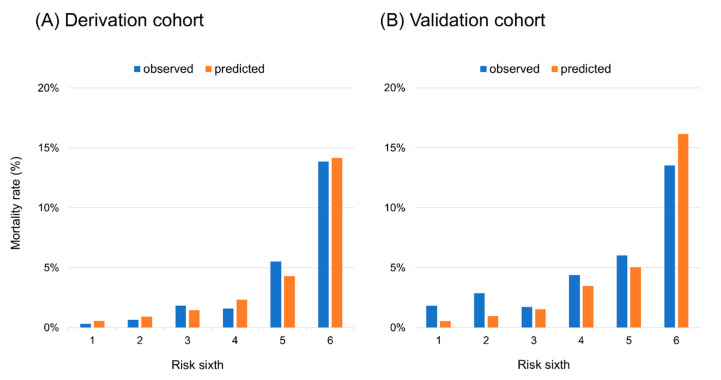
Calibration plots for the model applied to the (**A**) derivation cohort and (**B**) validation cohort.

**Figure 2 jcm-09-03394-f002:**
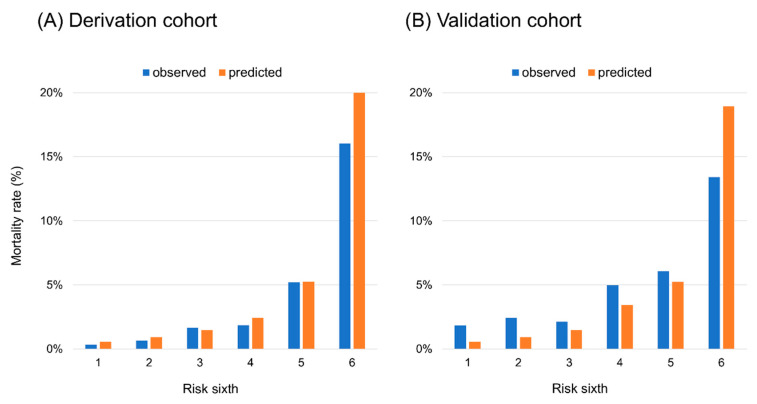
Calibration plots of the integer score applied to the (**A**) derivation cohort and (**B**) validation cohort.

**Table 1 jcm-09-03394-t001:** Baseline characteristics.

Variable	Derivation Cohort *n* = 4351	Validation Cohort *n* = 1682
Background		
Age, years	74.6 ± 13.1	77.5 ± 12.4 *
Male, %	59.9	55.4
Body mass index, kg/m^2^	23.2 ± 4.5	23.1 ± 4.8
Systolic blood pressure, mm Hg	140.1 ± 33.6	149.0 ± 37.1 *
Heart rate, beat per minute	93.7 ± 29.0	97.5 ± 28.3 *
Ejection fraction, %	43.5 ± 15.9	46.3 ± 16.0 *
NYHA functional class, %		
II	17.8	14.2 ^†^
III	37.8	37.5
IV	44.4	48.3 ^†^
Ischemic etiology, %	28.5	30.3
Comorbidities		
Prior admissions for heart failure, %	33.2	N/A
Hypertension, %	67.7	67.3
Hyperlipidemia, %	41.0	37.1 ^†^
Diabetes mellitus, %	35.1	36.9
Atrial fibrillation, %	47.6	39.1 *
Stroke, %	16.0	11.3 *
COPD, %	5.1	9.0 *
Laboratory findings at admission		
Hemoglobin, mg/dL	12.0 ± 2.3	11.7 ± 2.3 *
Creatinine, mg/dL	1.1 (0.8–1.5)	1.1 (0.8–1.6)
BUN, mg/dL	27.6 ± 17.3	29.5 ± 17.4 *
Sodium, mEq/L	139.2 ± 4.4	138.9 ± 4.7 ^†^
Total bilirubin, mg/dL	1.0 ± 0.7	0.9 ± 0.9 ^†^
Albumin, mg/dL	3.6 ± 0.5	3.5 ± 0.5 *
BNP, pg/mL ^‡^	658 (335–1209)	744 (444–1343) *
NT-proBNP, pg/mL ^‡^	3867 (1917–8741)	6820 (2908–13,840) *
Medication before admission		
Loop diuretics, %	46.1	50.9 ^†^
ACEI or ARB, %	44.1	46.2
Beta blocker, %	44.4	43.2
Aldosterone antagonist, %	18.4	22.1 ^†^
Digitalis, %	8.6	5.4 *
Treatment during admission		
Loop diuretics, IV, %	68.2	82.8 *
Vasodilator, IV, %	61.8	60.7
Inotrope, IV, %	16.1	16.1
Non-invasive ventilation, %	20.1	24.2 *
Intubation, %	4.4	6.7 *
IABP, %	1.9	1.6
VA-ECMO, %	0.5	0.4
Dialysis, %	4.7	3.3

Values are mean ± standard deviation, or median (interquartile range). Notes: * *p* value < 0.001; ^†^
*p* value < 0.01. ^‡^ In the derivation cohort, BNP and NT-proBNP levels were measured in 2597 and 1754 patients, respectively; in contrast, in the validation cohort, these levels were measured in 1481 and 201 patients, respectively. Abbreviations: COPD, chronic obstructive pulmonary disease; BUN, blood urea nitrogen; BNP, B-type natriuretic peptide; NT-proBNP, N-terminal pro-B-type natriuretic peptide; ACEI, angiotensin-converting enzyme inhibitor; ARB, angiotensin receptor blocker; IV, intravenous administration; IABP, intra-aortic balloon pump; VA-ECMO, venoarterial extracorporeal membrane oxygenation.

**Table 2 jcm-09-03394-t002:** Clinical variables contributing to in-hospital mortality.

Variable	Total/Missing Data (%)	Wald χ^2^	OR (95%CI)	*p* Value
Age (per 1 years)	4351/0 (100%)	22.49	1.04 (1.02, 1.05)	<0.0001
Male	4351/0 (100%)	0.75	0.87 (0.64, 1.19)	0.3852
Body mass index (per 1 kg/m^2^ increase)	4026/325 (92.5%)	35.19	0.87 (0.83, 0.91)	<0.0001
SBP (per 1 mm Hg increase)	4340/11 (99.7%)	44.40	0.98 (0.98, 0.99)	<0.0001
Heart rate (per 1 bpm increase)	4323/28 (99.4%)	2.89	1.00 (0.99, 1.00)	0.0890
LVEF (continuous)	4192/159 (96.3%)	7.85	0.98 (0.97, 1.00)	0.0051
LVEF (categorical)	4192/159 (96.3%)			
<20%			1.00 (reference)	
20−40%		13.42	0.36 (0.21, 0.62)	0.0002
>40%		20.04	0.30 (0.18, 0.51)	<0.0001
NYHA functional class	3974/377 (91.3%)			
II			1.00 (reference)	
III		7.30	3.28 (1.39, 7.76)	0.0069
IV		20.26	6.72 (2.93, 15.4)	<0.0001
Ischemic etiology	4346/5 (99.9%)	6.14	1.49 (1.09, 2.08)	0.0132
Prior admission for heart failure	4315/36 (99.2%)	25.24	2.21 (1.62, 3.01)	<0.0001
Diabetes mellitus	4345/6 (99.9%)	3.38	1.34 (0.98, 1.83)	0.0658
Atrial fibrillation	4337/14 (99.7%)	0.76	0.87 (0.63, 1.19)	0.3841
Stroke	4334/17 (99.6%)	2.83	1.39 (0.95, 2.03)	0.0925
COPD	4329/22 (99.5%)	6.37	1.99 (1.17, 3.39)	0.0116
Hemoglobin (per 1 mg/dL increase)	4345/6 (99.9%)	39.52	0.81 (0.75, 0.86)	<0.0001
GFR (per 1 mL/min/1.73 m^2^ increase)	4323/28 (99.4%)	38.40	0.98 (0.97, 0.98)	<0.0001
BUN (per 1 mg/dL increase)	4343/8 (99.9%)	118.25	1.03 (1.03, 1.04)	<0.0001
Total bilirubin (per 1 mg/dL increase)	4205/146 (96.6%)	17.40	1.35 (1.17, 1.55)	<0.0001
Albumin (per 1 mg/dL increase)	2834/1317 (65.1%)	95.06	0.22 (0.17, 0.30)	<0.0001
Hyponatremia (sodium ≤ 135 mEq/L)	4344/7 (99.9%)	45.62	3.23 (2.33, 4.55)	<0.0001
BNP >1000 pg/mLor NT-proBNP >4000 pg/mL	4219/132 (97.0%)	58.46	4.05 (2.83, 5.80)	<0.0001
Loop diuretics at baseline	4204/147 (96.6%)	18.21	2.01 (1.46, 2.76)	<0.0001
ACEI or ARB at baseline	4318/33 (99.2%)	4.37	0.71 (0.52, 0.98)	0.0366
Beta blockers at baseline	4313/38 (99.1%)	0.29	0.92 (0.67, 1.26)	0.5929
Aldosterone antagonists at baseline	4177/174 (96.0%)	5.86	1.58 (1.09, 2.28)	0.0155

OR, odds ratio; CI, confidence interval; SBP, systolic blood pressure; LVEF, left ventricular ejection fraction; NYHA, New York Heart Association; COPD, chronic obstructive pulmonary disease; GFR, glomerular filtration rate; BUN, blood urea nitrogen; BNP, B-type natriuretic peptide; NT-proBNP, N-terminal pro-B-type natriuretic peptide; ACEI, angiotensin-converting enzyme inhibitor; ARB, angiotensin receptor blocker.

**Table 3 jcm-09-03394-t003:** Multivariable logistic regression model predicting in-hospital mortality with multiple imputation by chained equations (the number of imputations: 10).

Variable	Beta Coefficient	OR	95% CI	*p* Value
Age				
<85 y/o		1.00	Reference	
≥85 y/o	0.772	2.16	(1.54, 3.04)	<0.0001
SBP				
>140 mm Hg		1.00	Reference	
100−140 mm Hg	0.496	1.64	(1.13, 2.39)	0.009
<100 mm Hg	1.386	4.00	(2.51, 6.36)	<0.0001
BUN				
<40 mg/dL		1.00	Reference	
≥40 mg/dL	0.889	2.43	(1.73, 3.43)	<0.0001
Sodium				
>135 mEq/L		1.00	Reference	
≤135 mEq/L	0.755	2.13	(1.46, 3.10)	<0.0001
Albumin				
>3.0 mg/dL		1.00	Reference	
≤3.0 mg/dL	1.428	4.17	(2.82, 6.16)	<0.0001
Natriuretic peptides				
BNP <1000 or NT-proBNP <4000 pg/mL		1.00	Reference	
BNP ≥1000 or NT-proBNP ≥4000 pg/mL	1.063	2.90	(1.97, 4.25)	<0.0001

Abbreviations: OR, odds ratio; CI, confidence interval; SBP, systolic blood pressure; BUN, blood urea nitrogen; BNP, B-type natriuretic peptide; NT-proBNP, N-terminal pro-B-type natriuretic peptide.

**Table 4 jcm-09-03394-t004:** A chart to calculate the integer risk score for each patient.

Variable	Risk Score				
Age (years)	<85	≥85			
	0	+2			
SBP (mmHg)	>140	100–140	<100		
	0	+1	+3		
BUN (mg/dL)	<40	≥40			
	0	+2			
Sodium (mEq/L)	>135	≤135		**Total** **score**	**Mortality rate**
	0	+2			
Albumin (mg/dL)	>3.0	≤3.0		0–1 2–4 5–6 7–8 9–14	<1% 1–5% 5–10% 10–30% >30%
	0	+3			
BNP or NT-proBNP (pg/mL)	<1000 or 4000	≥1000 or 4000			
	0	+2			
	**Total score =**				

**Table 5 jcm-09-03394-t005:** In-hospital treatment according to estimated risks by the integer score.

Variable	Low (Score 0–1)	Intermediate (Score 2–4)	High (Score 5–14)
Loop diuretics, IV, %	64.4	69.4	74.0 *
Vasodilators, IV, %	60.2	66.5	57.8 *
Inotropes, IV, %	6.9	14.9	27.7 *
Non-invasive ventilation, %	16.3	22.7	21.4 *
Intubation, %	3.1	4.7	4.8
IABP, %	0.8	2.3	3.1 *
VA-ECMO, %	0.3	0.6	0.6
Dialysis, %	0.6	4.6	5.8 *

Notes: * *p* value < 0.001. Abbreviations: IV, intravenous administration; IABP, intra-aortic balloon pump; VA-ECMO, venoarterial extracorporeal membrane oxygenation.

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
