# Peer review of "Derivation and Validation of Clinical Prediction Models for Rapid Risk Stratification for Time-Sensitive Management for Acute Heart Failure"

_jcm, 2020, doi:10.3390/jcm9113394_

Round 1

Reviewer 1 Report

The present manuscript, written by Yasuyuki Shiraishi and colleagues, is an original paper on the newly-developed predictive model for rapid risk stratification for acute heart failure (AHF). Briefly, the authors combined the impressive number of 6,033 patients form three multi-centre Japanese AHF registries, namely the WET-HF and NaDEF registries, that saved as a derivation cohort and the REALITY-AHF – used as a validation cohort. After applying appropriate sophisticated statistical methods, the authors developed risk model, consisting of six parameters of age, systolic blood pressure, blood urea, serum sodium, albumin, and natriuretic peptide that accurately predicted in-hospital mortality.

This is a well-planned, performed and written study. The applied methods are appropriate and generated results seems to be robust. The conclusions are based on the accumulated data. Nonetheless, there are several important concerns that need to be commented and perhaps corrected before the final disposition can be made.

Concerns:

  1. Based on the integer risk score, the authors created five sub-groups (quintals): (1) 0-1 with mortality rate < 1%; (2) 2-4 with mortality rate of 1-5%, etc. Yet, the authors tend to divide patients into sixtiles (Figure 1 and 2). This creates unnecessary confusion. It would be much better if the authors followed their own and logic differentiation into quintiles instead of sixitiles.
  2. The fact that only two-thirds of patients had albumin, that is one of key parameter of the model, measured is an important limitation. Obviously, the authors provided explanation and somehow assured that both the complete-case analysis model and the model derived from the imputation dataset provided similar results. However, generally speaking it is not a good practice to construct the model and include the parameter that is unavailable for one-third of the population.
  3. Moreover, there seems to be practical problems with the newly-created model just from the beginning. Although five predictors, such as age, systolic blood pressure, urea, sodium, and natriuretic peptides are typical parameters for all cardiac and many non-cardiac patients at the emergency settings – so called “must have”, it is the opposite for albumin. In real world, albumin is not a first and perhaps not even second-line laboratory measure for majority of cardiac patients. Thus, retrospective validation of this new model in other AHF cohorts (also outside Japan) will be problematic if not impossible. At the same time, prospective validation seems to be also questionable as currently there is not much data to recommend albumin screening for AHF patients. Although the cost of single albumin measurement is not high, nonetheless, incorporation of any new parameter into existing laboratory panels without clear data on its utility, will not be warmly welcomed by any hospital management, especially in the already strained hospital budgets. The worst case scenario may be the fact that this newly-developed model cannot be put into practice because of lack of key parameter and the whole impressive work done by the authors will be in vain. In order to find the solution – the authors hold extremely valuable datasets on several thousands of AHF patients. Did the authors explore all options to build such a model? Perhaps, some other widely available parameters, such as e.g. haemoglobin or ejection fraction may perform only marginally worse than albumin.
  4. The authors underline several times that management of AHF patients should be based on accurate risk assessment. It is hard to disagree. Thus, why the authors did not provide any data on the different (if any) management strategies according to their quintiles (or sixtiles)? If the authors want to convince the readers that incorporating this model into practice makes sense, they need to provide some proof. Further, it is somehow irritating – the authors write about low- and high-risk groups (presumably binary classification) and yet they divide patients into five or even six groups! Mortality of below 1% is low but how to interpret mortality of 8, 15, 24%? The more sub-groups (5 or 6) created, the more confusion regarding proper risk assessment and best therapy.
  5. Given the advanced age of the study populations, it is striking that ischemic aetiology of HF was present in less than one-third of patients. In Europe and US, ischemic heart disease is number one reason for HF in older patients. This important difference in HF aetiologies between Japanese and other countries may have huge implications as it is well-known and studied that ischemic HF has far worse prognosis than HF of other aetiologies. Consequently, the study results may not be applicable to HF population outside Japan.
  6. Overall, in-hospital mortality of 4% seems to low and even much lower than in other more general AHF registries. However, in-hospital mortality is only a fraction of the problem as generally in-hospital mortality is already relatively low in cardiac settings, i.e. acute coronary syndromes. Although not ideal (and there is much room for improvement), nevertheless, the applied therapies are successful in great majority of acute or sub-acute (NYHA class of II and III in almost half of studied patients). What is much more important at this stage is to accurately predict outcomes after hospital discharge as great majority of events, both deaths and urgent re-admissions occur afterwards. Do the authors have data on outcomes after hospital discharge as these would be equally (or even) more interested than those already provided?           

Reviewer 2 Report

The authors developed and validated a clinical prediction model for in-hospital mortality in patients with acute heart failure (AHF). The present study analyzed using data from three AHF registry cohorts. This prediction model including 6 clinical variables is easy to use and provided good discrimination and calibration for in-hospital mortality.

The paper is well written, statistics and summary data are clear. However, there seem to be problems requiring further discussion.

Major comments:

  1. As stated in the discussion section, natriuretic peptides and nutritional status are prognostic indicators for patients with HF, and the risk model the authors developed in the present study included BNP or NT-proBNP and serum albumin. On the other hand, the authors validated the GWTG-HF risk score, including variables overlapped with this study, in Japanese AHF patients, and described the improvement of its discrimination by adding B-type natriuretic peptide level [Am Heart J. 2016 Jan;171(1):33-9]. Therefore, we recommend that the authors compare the difference between the two models' discriminative ability and describe the present study's superiority.
  2. According to clinical variables shown in Table 2, most variables such as age and BMI are treated as continuous variables, while LVEF and BNP/NT-proBNP are treated as categorical variables, not as continuous variables. Please explain the rationale of the cut-off points using in this analysis.
  3. In the results section, the authors state that no other differences were evident between the derivation and validation cohorts regarding the baseline characteristics except for age, SBP, LVEF, and BNP or NT-proBNP levels. However, we think that the patients in the validation cohort tended to be more likely to also have a stroke, COPD, etc. P value was not shown in Table 1, and the definition of the word "tended to" is unclear. Please describe the difference between the derivation and validation cohorts in more detail.
  4. Methods section of the main paper: Provide the dates of this analysis given the age of the data.
